# DIME: An Information-Theoretic Difficulty Measure for AI Datasets

## Abstract

Evaluating the relative difficulty of widely-used benchmark datasets across time and across data modalities is important for accurately measuring progress in machine learning. To help tackle this problem, we propose **DIME**, an information-theoretic **DI**fficulty **ME**asure for datasets, based on conditional entropy estimation of the sample-label distribution. Theoretically, we prove a model-agnostic and modality-agnostic lower bound on the 0-1 error by extending Fano's inequality to the common supervised learning scenario where labels are discrete and features are continuous. Empirically, we estimate this lower bound using a neural network to compute DIME. DIME can be decomposed into components attributable to the data distribution and the number of samples. DIME can also compute per-class difficulty scores. Through extensive experiments on both vision and language datasets, we show that DIME is well-aligned with empirically observed performance of state-of-the-art machine learning models. We hope that DIME can aid future dataset design and model-training strategies.

## 1 Introduction

Empirical machine learning research relies heavily on comparing performance of algorithms on a few standard benchmark datasets. Moreover, researchers frequently introduce new datasets that they believe to be more challenging than existing benchmarks. However, we lack objective measures of dataset difficulty that are independent of the choices made about algorithm- and model-design. Moreover, it is also hard to compare algorithmic progress across data modalities, such as language and vision. So, for instance, it is difficult to compare the relative progress made on a sentiment analysis benchmark such as the Stanford Sentiment Treebank (SST) (Socher et al., 2013) and an image classification benchmark, like CIFAR-10 (Krizhevsky, 2009). With these challenges in mind, we propose a model-agnostic and modality-agnostic measure for comparing how difficult it is to perform supervised learning on a given dataset.

Intuitively, assuming that dataset examples are sampled i.i.d. from a static true distribution, we argue that the difficulty of a dataset can be decoupled into two relatively independent sources:(a) approximation complexity, the number of samples required to approximate the true distribution up to certain accuracy, and (b) distributional complexity, the intrinsic difficulty involved in modeling the statistical relationship between the labels and features.

We focus our analysis on the second source of the intrinsic difficulty in supervised learning, where both features and labels are available. To provide a model-agnostic measure, we turn to the information-theoretic approach. Indeed, there already exist lower bounds on the lowest possible errors given the distribution of the data. If both the samples and labels are discrete, Fano's inequality suggests the probability of 0-1 error is bounded by terms related to the conditional entropy $\mathbb{H}(Y|X)$, where $X$ is the random variable representing the features and $Y$ is the label. When both the features $X$ and label $Y$ are continuous, results on differential entropy also suggest the expected $L_2$ error is lower-bounded. However, in most of the supervised learning datasets where the features are continuous and the labels are discrete, it is unknown how the lowest possible error can be controlled regardless of models.

In this paper, we show that even for the hybrid case where labels are discrete and features are continuous, with some additional assumptions, Fano's inequality still holds. Moreover, we show that the lowest possible probability $\mathcal{P}_e$ of the 0-1 error for a given data distribution is lower bounded

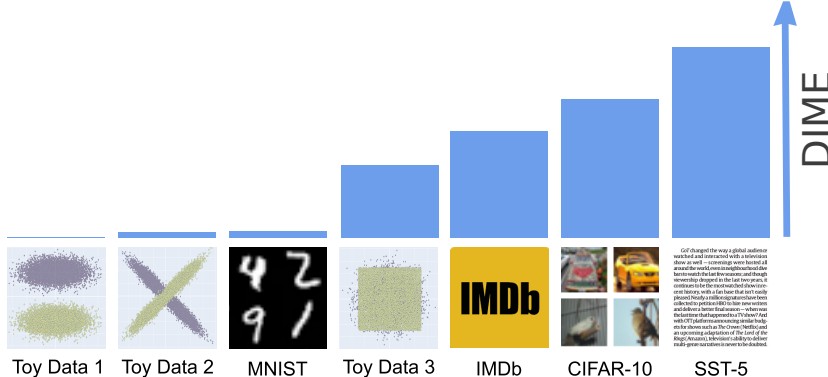

Figure 1: We propose DIME, a model- and modality-agnostic difficulty metric for datasets

by terms related to a hybrid conditional entropy $\mathbb{H}(Y|X)$. We further design an estimator for the lower bound of $\mathcal{P}_e$ based on our generalized Fano's inequality. The estimator uses neural networks to approximate the KL divergence based on Donsker-Varadhan representation, which is then used to estimate the hybrid conditional entropy $\mathbb{H}(Y|X)$ as well as the lower bound of $\mathcal{P}_e$.

We emphasize that even though our lower bound is model-agnostic, the proposed estimator is based on a neural network. However, we empirically show that, for most image and natural language datasets, a multilayer perceptron-based estimator produces a measure that effectively captures the difficulty of the data and aligns well with the performance of state-of-the-art models.

**Related Work:** Although conditional entropy and mutual information estimation have been extensively studied, research has focused on purely discrete or continuous data. Nair et al. (2006) were among the first to study the theoretical properties for the case of mixture of discrete and continuous variables. Ross (2014), Gao et al. (2017) and Beknazaryan et al. (2019) proposed approaches for estimating mutual information for the mixture case based on density ratio estimation (e.g.,binning, kNN or kernel methods), which is unsatisfactory for high dimensional data such as image and text. We use neural network estimation (Belghazi et al., 2018) to avoid these issues. More importantly, we are the first to connect the hybrid conditional entropy with the lowest classification error and are able to use it as a difficulty measure for datasets.

## 2 DESIGNING A DATASET DIFFICULTY MEASURE

For supervised learning across data modalities such as images and text, data samples can usually be viewed as feature-label pairs $(x, y)$ where $x \in \mathcal{X} \subset \mathbb{R}^{d_x}$, and $y \in \mathcal{Y}$. We focus on classification problems where the labels $y$ are discrete, i.e., $\mathcal{Y} \subset \mathbb{Z}^+$. We denote the joint distribution of the feature-label pairs as $\mathcal{P}_{\mathcal{X}\mathcal{Y}}$. The marginal distributions of the features and labels are denoted as $\mathcal{P}_{\mathcal{X}}$ and $\mathcal{P}_{\mathcal{Y}}$ respectively. We make the following widely adopted assumption from learning theory literature about how samples are generated:

**Assumption 1.** *The feature-label pairs $(x, y)$ in the datasets, both training and testing, are sampled i.i.d. from a static distribution $(x, y) \sim \mathcal{P}_{\mathcal{X}\mathcal{Y}}$.*

Intuitively, there are many possible indicators for the potential difficulty of a dataset: the number of features, the number of classes, the number of samples, the distinguishability of samples across classes, as well as the difference between the data distributions of the training set and the testing set. However, none of these indicators alone can fully describe the relative difficulty of a dataset.

From Assumption 1, if the samples $(x, y)$ are sampled i.i.d. from $\mathcal{P}_{\mathcal{X}\mathcal{Y}}$, where $\mathcal{Y}$ is discrete, a natural measure that characterizes the difficulty of the data distribution is the best probability of the 0-1 error that can be achieved by any estimator.

**Definition 1** (Model-Agnostic Error). $\mathcal{P}_e = \inf_f \mathcal{P}_{x,y \sim \mathcal{P}_{\mathcal{X}\mathcal{Y}}} [f(x) \neq y]$

The measure 1 is straightforward, but unfortunately it is hard to compute since it involves evaluations against all possible estimators. However, with mild assumptions, it can be lower bounded by terms related to the conditional entropy, which is much easier to evaluate.

## 2.1 DISCRETE FEATURES

In the case where the features $x \in \mathcal{X}$ are discrete, according to Fano's theorem, $\mathcal{P}_e$ is lower bounded:

**Fano's inequality.** *If both $X$ and $Y$ are discrete random variables, then $\mathcal{P}_e \geq \frac{\mathbb{H}(Y|X)-1}{\log |\mathcal{Y}|}$, where $|\mathcal{Y}|$ is the cardinality of the label set $\mathcal{Y}$, and $\mathbb{H}(Y|X)$ is the conditional entropy.*

However, even though data can be represented using discrete integers, treating the features as discrete random variables leads to the following difficulties:

1. The cardinality of the feature space becomes extremely large if discrete features are used. For image data, since each pixel is represented as an integer, the (raw) feature dimension would become the number of pixels in the image, which can be extremely large. Similarly, for language data, when the sequence length is long, the feature dimension becomes large very quickly.

2. Given the large feature space, finding a matching set of features between training and testing data from the limited number of training and test samples would be unlikely. As a result, probability mass estimation on each discrete value would be impractical, since we may only see at most one sample for each discrete configuration.

## 2.2 CONTINUOUS FEATURES

As opposed to treating the features $x$ as discrete random variables, if we view them as i.i.d. samples from a continuous distribution with probability density $p_x$, we can estimate the conditional entropy $\mathbb{H}(Y|X)$ under some smoothness assumptions and also infer the model-agnostic error $\mathcal{P}_e$.

However, classical Fano's inequality only holds for discrete random variables. For the case with the continuous features and discrete labels, it has not been shown how $\mathcal{P}_e$ can be controlled. In this paper, we prove a generalized version of Fano's inequality that holds for the continuous-feature-discrete-label scenario. Formally, for the continuous-$X$-discrete-$Y$ case:

**Definition 2.** *Hybrid Conditional Entropy*

$$\mathbb{H}(Y|X) := \mathbb{E}_X \left[ -\sum_{y=1}^{|\mathcal{Y}|} \mathbb{P}(Y = y|X) \log \mathbb{P}(Y = y|X) \right], \tag{1}$$

*where $\mathbb{P}(Y = y|X) := \mathbb{E}[\mathbf{1}(Y = y)|X]$.*

This definition of $\mathbb{H}(Y|X)$ is consistent with the classical definition, in the sense that both of them give the intuition that how much information or uncertainty is left for $Y$ given $X$.

Next, to connect $\mathcal{P}_e$ and the hybrid $\mathbb{H}(Y|X)$, we introduce an assumption on the function $f$

**Definition 3.** *Smooth Discretization Property: The function $f : \mathcal{X} \to \mathcal{Y}$ satisfies the smooth discretization property if for every $y \in \mathcal{Y}$, almost every $x \in \mathcal{X}$ ($x$ a.e. in $\mathcal{X}$),*

$$f(x) = y \iff \exists \delta > 0 \quad s.t. \quad \forall \tilde{x} \in B_\delta(x), \quad f(\tilde{x}) = y,$$

*where $B_\delta(x) := \{\tilde{x} \in \Omega : \|\tilde{x} - x\|_2 < \delta\}$ is a $\delta-$neighborhood of $x$ in $\mathcal{X}$.*

This assumption on the classifier function $f$ is not unnatural considering $f$ maps a continuous variable to a discrete variable. Without this assumption, it would be extremely hard to quantify the population error probability $P(f(X) \neq Y)$ since the behavior of $f$ may be erratic. Furthermore, in the real data setting, this assumption is always satisfied for every classifier $f$, as we can always construct a small enough neighborhood of each data point such that they are disjoint and assume $f$ is a constant in each neighborhood. In this sense the assumption on $f$ is pretty minimal. Now we are ready to extend Fano's inequality:

**Theorem 1.** *Fano's Inequality for Continuous Features: Let $\mathcal{P}_e$ be the minimum error probability, i.e.,*

$$\mathcal{P}_e = \inf_f P(f(X) \neq Y)$$

*where $f$ is any estimator of $Y$ based on the observation $X$ that satisfies the **smooth discretization property**. Then we have*

$$\mathbb{H}(\mathcal{P}_e) + \mathcal{P}_e \log(m - 1) \geq \mathbb{H}(Y|X), \tag{2}$$

*where $\mathbb{H}(\mathcal{P}_e) := -\mathcal{P}_e \log \mathcal{P}_e - (1 - \mathcal{P}_e) \log(1 - \mathcal{P}_e)$.*

*Proof.* See appendix B. $\square$

## 3 ESTIMATING THE LOWER BOUND

It is natural to consider $\mathcal{P}_e$ defined in Theorem 1 as a measure of dataset difficulty. Unfortunately, direct estimation of $\mathcal{P}_e$ is impractical since one has to evaluate the estimation error against all possible estimators. However, theorem 1 provides an alternative towards estimating a lower bound on $\mathcal{P}_e$ through estimating the hybrid conditional entropy $\mathbb{H}(Y|X)$ defined in Equation (1).

### 3.1 CONDITIONAL ENTROPY ESTIMATION

In real applications, direct calculation of hybrid conditional entropy $\mathbb{H}(Y|X)$ according to Definition 2 is impossible since $\mathbb{P}(Y = y|X)$ is unknown. However, similar to the conditional entropy for discrete random variables, the hybrid conditional entropy $\mathbb{H}(Y|X)$ can also be written as

$$\mathbb{H}(Y|X) = \mathbb{H}(Y) - \sum_{y=1}^{|\mathcal{Y}|} P(Y = y)\mathbb{KL}(X|Y = y||X). \tag{3}$$

Please refer to Appendix A for a detailed proof of Equation (3). We also define the hybrid mutual information $\mathbb{I}(X; Y)$, which is compatible with $\mathbb{H}(Y|X)$:

$$\mathbb{I}(X; Y) = \sum_{y=1}^{|\mathcal{Y}|} P(Y = y)\mathbb{KL}(X|Y = y||X) \tag{4}$$

In some benchmark datasets with balanced classes (e.g., CIFAR-10 and MNIST), $\mathbb{H}(Y|X) = \log|\mathcal{Y}| - \frac{1}{|\mathcal{Y}|} \sum_{y=1}^{|\mathcal{Y}|} \mathbb{KL}(X|Y = y||X)$. This indicates that if a dataset has more classes and the features for different classes are closer to each other on average, then $\mathbb{H}(Y|X)$ would be larger.

### 3.2 A VARIATIONAL KL DIVERGENCE ESTIMATOR USING NEURAL NETWORKS

We now discuss the practical design of an estimator for $\mathbb{KL}(X|Y)$. Even though there are non-parametric KL divergence estimators such as $k$NN based techniques (Noshad et al., 2017; Perez-Cruz, 2008; Wang et al., 2009), they rely on some knowledge of the internal dimension of the data manifold. However, the internal dimension of the data manifold is usually much smaller than the dimension of the raw features and is hard to estimate. As an alternative, we design a neural network-based estimator for KL divergence estimation, inspired by Belghazi et al. (2018). The estimator is based on the following theorem:

**Donsker-Varadhan representation.**

$$\mathbb{KL}(\mathcal{P}||\mathcal{Q}) = \sup_{T:\mathcal{X}\to\mathbb{R}} \mathbb{E}_{\mathcal{P}}[T] - \log \mathbb{E}_{\mathcal{Q}}[e^T] \tag{5}$$

If we parameterize the function $T$ using a neural network, we obtain:

$$\mathbb{KL}(\mathcal{P}||\mathcal{Q}) \geq \sup_{\theta\in\Theta} \mathbb{E}_{\mathcal{P}}[T_\theta] - \log \mathbb{E}_{\mathcal{Q}}[e^{T_\theta}], \tag{6}$$

where $\theta$ represents the neural network parameters.

In our empirical evaluation, we optimize the empirical average instead of the expectation. However this may cause issues such as overfitting. To mitigate this issue, we split the data into separate training and validation sets. The neural network models are trained on the training set and the estimation is made using the validation set. See Algorithm 1 for details.

---

**Algorithm 1** DIME

---

Initialize $|\mathcal{Y}|$ neural networks: $T_{\theta_1}, T_{\theta_2}, \ldots, T_{\theta_{|\mathcal{Y}|}}$, one per class.
Initialize class counters $k_1 = 0, \ldots, k_c = 0$, one per class. Initialize the sample counter $k = 0$.
**for** $t$ in $0, \ldots, T$ **do**
    Draw a batch of $b$ training examples $\{(x_i, y_i)\}$, and $b$ evaluation examples $\{(x_i^e, y_i^e)\}$.
    **for** each class $c \in \{1, \ldots, |\mathcal{Y}|\}$ **do**
        Find samples of the $c$-th class: $S_c = \{i|y_i = c\}$, $S_c^e = \{i|y_i^e = c\}$
        Train $\theta_c = \arg\max_{\theta_c} \frac{1}{|S_c|} \sum_{S_c} T_{\theta_c}(x_i) - \log[\frac{1}{b} \sum_j \exp T_{\theta_c}(x_j)]$
        Evaluate $\widehat{\mathbb{KL}}_{\theta_c} = \frac{1}{|S_c^e|} \sum_{S_c^e} T_{\theta_c}(x_i^e) - \log[\frac{1}{b} \sum_j \exp T_{\theta_c}(x_j^e)]$
        If $\widehat{\mathbb{KL}}_{\theta_c}$ stops increasing, pause the training for $T_{\theta_c}$.
        Update counters: $k_c + = |S_c^e|$.
    **end for**
    Update the sample counter: $k + = b$
    If all neural networks stop updating, break.
**end for**
Estimate the class probability $\widehat{p}_c = \frac{k_c}{k}$, $c = \{1, \ldots, |\mathcal{Y}|\}$.
Calculate $\widehat{\mathbb{H}}(Y|X)$ using (3), $\widehat{p}_c$, and $\widehat{\mathbb{KL}}_{\theta_c}$, $c = \{1, \ldots, |\mathcal{Y}|\}$.
Solve (2) and output the solution.

---

### 3.3 DIME AS A DIFFICULTY MEASURE

Given that we are using a neural network model to estimate the KL divergence between $\mathcal{P}(X|y)$ and $\mathcal{P}(X)$, according to Equation (6), $\widehat{\mathbb{KL}}(X|y||X) \leq \mathbb{KL}(X|y||X)$. As a consequence, the estimated mutual information $\widehat{\mathbb{I}}(X;Y) \leq \mathbb{I}(X;Y)$, which leads to a larger $\widehat{\mathbb{H}}(Y|X)$. In the end, DIME could be larger as compared to true lower bound on $\mathcal{P}_e$.

The caveat is that $\mathcal{P}_e$ is defined as a model-agnostic measure, but we use a neural network model to estimate its lower bound instead. The hope is, if the function class of the neural network is large enough, the gap between $\widehat{\mathbb{KL}}(X|y||X)$ and $\mathbb{KL}(X|y||X)$ is small enough so that DIME won't deviate too much from the true lower bound of $\mathcal{P}_e$. We now describe our experiments for DIME using vision and language benchmark datasets.

## 4 EXPERIMENTS AND RESULTS

We try to answer the following questions through our experiments: Is the neural net estimator tight-enough? How does DIME align with the state-of-the-art benchmarks across modalities? What are the practical applications for DIME in evaluating machine learning progress?

### 4.1 DATA PREPARATION

We evaluate the following image classification datasets: MNIST (LeCun et al., 1998), Extended MNIST (EMNIST) (Cohen et al., 2017), Fashion MNIST (Xiao et al., 2017a), CIFAR-10 (Krizhevsky, 2009), CIFAR-100 (Krizhevsky, 2009), Tiny ImageNet[1], and SVHN (Netzer et al., 2011). The EMNIST dataset has several different splits, which include splits by digits, letters, merge, class, and balanced. For a controlled evaluation study, we also add the PyTorch Fakedata dataset[2] which contains 10-classes of randomly-generated $32 \times 32$ grayscale images.

Among natural language processing datasets, we investigate three supervised scenarios: sentiment analysis, text classification, and language modeling. For sentiment analysis, we analyze IMDb (Maas et al., 2011), and Stanford Sentiment Treebank (SST) (Socher et al., 2013). In the case of SST, we utilize both fine-grained labels (SST-5) and binary labels (SST-2). For the text classification task, we use the TREC dataset (Li & Roth, 2002), AG News, DBPedia, as well as Yelp Review dataset (Zhang et al., 2015). For language modeling task, we analyze the character level modeling for Penn Tree Bank (Marcus et al., 1994).

---

[1]https://tiny-imagenet.herokuapp.com/
[2]https://github.com/pytorch/vision/blob/master/torchvision/datasets/fakedata.py

For DIME estimation, we do not perform any preprocessing on the data except for simple rescaling. For image data we rescale all the pixel values to floats in $[0, 1]$. By data processing inequality (Cover & Thomas, 2012), the rescaling will not cause any change on $\mathbb{H}(Y|X)$ and $\mathcal{P}_e$ since the rescaling function is invertible.

## 4.2 CHOICE OF NEURAL NETWORK ESTIMATOR

To estimate DIME for image datasets, we use a simple multi-layer perceptron (MLP) with residual layers with Rectified Linear Unit (ReLU) activation functions. The network has three hidden layers of 4096 neurons. For language datasets, we use an embedding layer of dimension 1500, followed by three layers of 256 hidden neurons with residual connections and ReLU activations. We found that a higher dimension of the embedding helps achieve a tighter bound. The embedding layer is initialized with a concatenation of five pretrained embedding vectors: GloVe-840B-300d, GloVe-42B-300d, GloVe-twitter-27B-200d, GloVe-6B-300d, FastText-en-300d, and CharNGram-100d. The input sequences are truncated or padded to the length of 128. The MLP is operating on the token dimension instead of the embedding dimension to take the order of the sequence into consideration. For example, the first layer of MLP is 128-by-256 instead of 1500-by-256. The last fully-connected layer flattens everything into a vector and projects it to a scalar. For optimizing the MLP, we use SGD with initial learning rate of 0.1 and anneal it to 0.01 and 0.001 if the objective stops updating. We use the test set as evaluation set, as described in Algorithm 1.

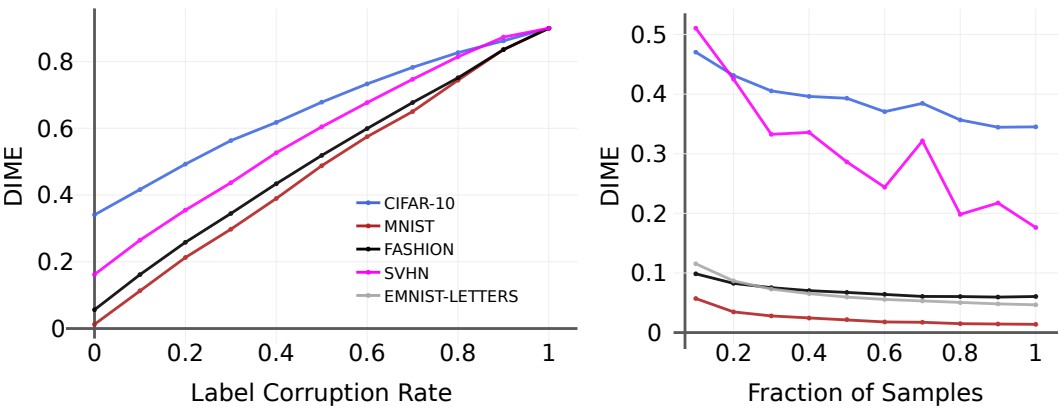

Figure 2: DIME increases with increase in the amount of label corruption (left). DIME becomes reasonably stable as the fraction of samples from the dataset used for estimation increase (right).

## 4.3 SANITY-CHECK EXPERIMENTS

**Experiments with toy datasets:** We generate three two-dimensional 2-class toy datasets with 20,000 samples each with increasing level of difficulty (Figure 1). Samples in Toy Data 1 and Toy Data 2 are generated using random Gaussian variables by scaling differently in different dimensions and then by rotating, while the two classes in Toy Data 3 are simply uniform and Gaussian random variables. Toy Data 1, which is relatively easy to be separated, gets a DIME of 0.001. Toy Data 2, which is a bit harder due to a small overlapping region around the axis origin, gets a DIME of 0.013. And Toy Data 3, which is the hardest due to overlapping supports of the two class, gets a DIME of 0.263. In sum, DIME accurately reflects the relative difficulty of the toy examples.

**Label corruption test:** Next, we estimate DIME for the case of label corruption. If we assign random labels to a fraction of the training samples, intuitively the dataset should become more difficult. We experiment with five datasets with ten classes each—MNIST, Fashion MNIST, CIFAR-10, EMNIST (digits), and SVHN—and perform label corruption on their training sets in steps of 0.1 from 0 to 1. We observe a consistent increase in DIME across all the datasets with increasing label corruption (Figure 2-(left)). For some datasets such as MNIST, DIME is almost tight given the fraction of randomly corrupted labels.

| Corpus | #classes | $\widehat{\mathbb{H}}(Y)$ | $\widehat{\mathbb{I}}(X;Y)$ | $\widehat{\mathbb{H}}(Y\|X)$ | DIME | SOTA Error |
|---|---|---|---|---|---|---|
| **Image Classification** | | | | | | |
| EMNIST (digits) | 10 | 2.303 | 2.255 | 0.048 | 0.006 | 0.002 |
| MNIST | 10 | 2.301 | 2.192 | 0.109 | 0.015 | 0.002 |
| EMNIST (letters) | 26 | 3.258 | 2.872 | 0.386 | 0.054 | 0.056 |
| EMNIST (bymerge) | 47 | 3.554 | 3.098 | 0.456 | 0.060 | 0.190 |
| Fashion-MNIST | 10 | 2.303 | 1.912 | 0.391 | 0.066 | 0.033 |
| EMNIST (byclass) | 62 | 3.679 | 3.126 | 0.553 | 0.072 | 0.240 |
| EMNIST (balanced) | 47 | 3.850 | 3.212 | 0.638 | 0.089 | 0.095 |
| SVHN | 10 | 2.223 | 1.436 | 0.786 | 0.159 | 0.010 |
| CIFAR-10 | 10 | 2.303 | 0.915 | 1.388 | 0.340 | 0.010 |
| CIFAR-100-Subclass | 10 | 2.303 | 0.503 | 1.800 | 0.504 | N/A |
| CIFAR-100 | 100 | 4.605 | 1.239 | 3.366 | 0.585 | 0.087 |
| Tiny ImageNet | 200 | 5.298 | 0.692 | 4.606 | 0.768 | 0.268 |
| FakeData | 10 | 2.303 | -0.003 | 2.306 | 0.900 | 0.900 |
| **Sentiment Analysis** | | | | | | |
| IMDb | 2 | 0.693 | 0.224 | 0.469 | 0.178 | 0.038 |
| SST-2 | 2 | 0.693 | 0.224 | 0.469 | 0.179 | 0.032 |
| SST-5 | 5 | 1.573 | 0.230 | 1.342 | 0.470 | 0.356 |
| **Text Classification** | | | | | | |
| DBPedia | 14 | 2.639 | 2.387 | 0.252 | 0.037 | 0.013 |
| TREC | 6 | 1.638 | 1.185 | 0.453 | 0.092 | 0.019 |
| YelpReview (Polarity) | 2 | 0.693 | 0.371 | 0.322 | 0.098 | 0.044 |
| AG News | 4 | 1.386 | 0.808 | 0.579 | 0.147 | 0.076 |
| **Language Modeling** | | | | | | |
| Penn Treebank | 42 | 2.966 | 1.876 | 1.090 | 0.165 | 1.083 (ppl) |

Table 1: We evaluate DIME on vision and language datasets and rank them by relative difficulty. Comparisons with prediction performance of state-of-the-art neural network models shows that DIME is roughly aligned with empirically observed performance. (ppl: perplexity)

### 4.4 EVALUATING BENCHMARK DATASETS IN VISION AND LANGUAGE

We evaluate DIME using popular image and language domain datasets for various classification tasks. In addition to dataset statistics, we report the estimated label entropy $\widehat{\mathbb{Y}}$, the estimated hybrid mutual entropy (Eq. 4), the estimated hybrid conditional entropy $\widehat{\mathbb{H}}(Y|X)$ (Eq. 1), and DIME. We also report the state-of-the-art error on the dataset from recent literature (Wan et al., 2013; Huang et al., 2018; Cubuk et al., 2019; Yang et al., 2019; Patro et al., 2018; Cer et al., 2018; Melis et al., 2019). From the results summarized in Table 1, we make the following empirical observations:

First, DIME align well with the state-of-the-art results on the benchmark datasets, indicating that the relative difficulty of datasets is reflected in the performance of latest machine learning algorithms. However, notably, DIME is lower than the reported state-of-the-art error for all the variants of EM-NIST. This could indicate that since EMNIST is a relatively new, less-investigated dataset, we might see model performance improving over time.

Second, a larger number of classes lead to a higher value of DIME in general, which is expected given its dependence on $\widehat{\mathbb{H}}(Y)$. But $\widehat{\mathbb{H}}(Y)$ does not always dominate the value of the measure. For example, MNIST has 10 classes, but its DIME is much smaller compared to IMDb which has only two classes. This relative difference in difficulty of MNIST and IMDb is also reflected in their SOTA errors, validating our measure.

Third, we can also evaluate the relative difficulty of similar datasets introduced over time. For instance, Fashion-MNIST, which was designed to be more challenging than MNIST (Xiao et al., 2017b), can be compared with MNIST on the basis of DIME. We find that Fashion MNIST is indeed more difficult than MNIST, though not as difficult as SVHN, indicating that with more exploration, we might see SOTA error approaching zero on Fashion MNIST.

Finally, for datasets such as MNIST, Fashion MNIST, EMNIST, FakeData, and DBPedia, DIME provides a tight error bound. But for certain datasets such as CIFAR-10, CIFAR-100, and Tiny ImageNet, DIME seem slightly pessimistic. As we discussed in Section 3.3, DIME could be larger than the true lower bound of $\mathcal{P}_e$ since we constrain the function space to be a neural network model. While the relative order of difficulty suggested by DIME is reasonable, its large value also suggests

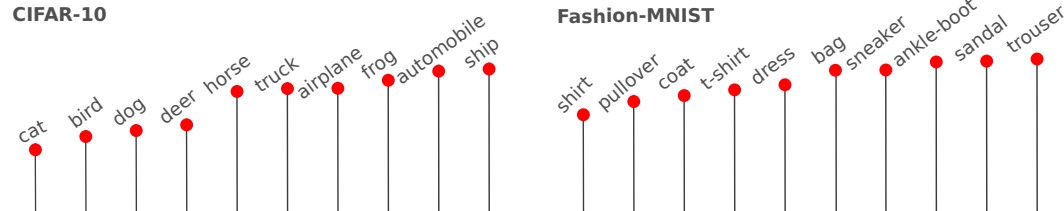

Figure 3: DIME can rank classes within datasets for difficulty. A lower height of the bar indicates higher relative difficulty.

than an MLP may not have enough capacity for accurate $P_e$ estimation. While it is an open question how to choose a model that is large enough to approximate $\mathcal{P}_e$ yet easy to optimize, Equation (6) suggests that as the model size grows, the gap between DIME and true lower bound of $\mathcal{P}_e$ becomes smaller. We verify this by calculating DIME with increasing number of neurons for a three-layer MLP. See Figure 5 in Appendix C for details.

### 4.5 ADDITIONAL USE CASES FOR DIME

**Dataset difficulty per class:** Algorithm 1 can also estimate the relative difficulty of classes within a dataset. For this purpose, we simply rank the classes by their $\widehat{\mathbb{KL}}(X|y||X)$. A smaller value of $\widehat{\mathbb{KL}}(X|y||X)$ indicates the class is harder to classify. As examples in Figure 3 show, 'cat' and 'bird' are the most difficult classes in CIFAR-10, while 'automobile' and 'ship' are the easiest. In addition, 'shirt' and 'pullover' are the most difficult classes in Fashion MNIST, while 'sandal' and 'trouser' are the easiest.

**Controlled Study with CIFAR Subsets:** We obtain a 10-class subset from CIFAR-100 by choosing subclasses from the 'large carnivores' and 'large omnivores and herbivores' superclasses. These classes—bear, leopard, lion, tiger, wolf, camel, cattle, chimpanzee, elephant, and kangaroo—should be intuitively harder to classify as compared to CIFAR-10, which includes disparate classes such as bird, truck, and ship. Note that the image data for both of these datasets comes from the same, larger '80 million tiny images' dataset (Torralba et al., 2008). We find that the DIME for this 10-class CIFAR-100 subset is 0.504, significantly larger than a DIME of 0.340 for CIFAR-10.

**Effect of number of samples:** With increasing number of samples, the empirical distribution of both training and testing tend towards the population distribution, making the distance between them smaller. It should also make the dataset easier, which is indeed reflected in a decreasing value of DIME with increasing number of samples (Figure 2-(right)). But we also observe that our measure becomes stable with increasing number of samples. This suggests that our method can be used to estimate the relative difficulty of new datasets in domains where data is hard to collect (e.g., medical datasets, fine-grained image classification) and make determinations if additional data collection is required to make the problem easier to solve.

**Effect of model size:** We investigate the effect of model size on DIME optimization. Equation (6) suggests that as the estimator model becomes larger, the gap between DIME and the true lower bound on $\mathcal{P}_e$ should become smaller. We experiment with different sizes of MLP and find that DIME generally decreases with increasing number of neurons. However for easier datasets such as MNIST and Fashion MNIST, DIME stabilizes fairly quickly. See Figure 5 from Appendix C for details.

## 5 CONCLUSION

We extend Fano's inequality to the case of continuous features and discrete labels and prove a model- and modality-agnostic lower bound on the 0-1 errors. We further design DIME, an empirical difficulty measure for datasets. We note that, even though our lower bound is model-agnostic, our estimator is based on a neural network and the estimates are affected by choices about the neural network design. Designing better estimators remains an avenue for future exploration. However, we hope that this work can aid dataset-design in the future and help objectively compare the progress in machine learning algorithms across modalities. We will release code for DIME estimation and DIME estimates for many common computer vision and natural language datasets to help further research.

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

## A    Hybrid conditional entropy

Consider the usual classification setting, $X$, representing the feature, is a random vector in $\mathbb{R}^p$; $Y$, representing the label, is a random variable(or vector) in $\{1, \ldots, |\mathcal{Y}|\}$. Assume $X$ is a continuous variable with density $p_X(x)$ for $x$ in some compact domain $\Omega \subset \mathcal{X} \subset \mathbb{R}^p$. Can we define a hybrid conditional entropy $\mathbb{H}(Y|X)$ while $X$ is continuous and $Y$ is discrete? We give a formal definition for this mixed-pair case and demonstrate that $\mathbb{H}(Y|X)$ defined in our way can be a good indicator for the hardness of classification on the data $(X, Y)$, which is consistent with the classical definition in the sense that both of them give the intuition that how much information or uncertainty is left for $Y$ given $X$.

Our definition is

$$\mathbb{H}(Y|X) := \mathbb{E}_X \left[ -\sum_{y=1}^{|\mathcal{Y}|} \mathbb{P}(Y = y|X) \log \mathbb{P}(Y = y|X) \right], \tag{7}$$

where $\mathbb{P}(Y = y|X) := \mathbb{E}[\mathbf{1}(Y = y)|X]$. By the definition of conditional expectation (or probability), it is easy to check that

$$\mathbb{E}[\mathbf{1}(Y = y)|X] = \frac{P(Y = y)p(X|Y = y)}{p_X(X)}, \tag{8}$$

where $p(x|Y = y), x \in \mathbb{R}^p$ is the conditional density function of $X$ given $Y = y$. Denote $p_y := P(Y = y)$, we have

$$\mathbb{H}(Y|X) = \int_\Omega -\sum_{y=1}^{|\mathcal{Y}|} p_y p(x|Y = y) \log \frac{p_y p(x|Y = y)}{p_X(x)} dx$$

$$= -\sum_{y=1}^{|\mathcal{Y}|} p_y \log p_y - \sum_{y=1}^{|\mathcal{Y}|} p_y \int_\Omega p(x|Y = y) \log \frac{p(x|Y = y)}{p_X(x)} dx$$

$$= \mathbb{H}(Y) - \sum_{y=1}^{|\mathcal{Y}|} p_y \mathbb{KL}(X|Y = y||X). \tag{9}$$

This proves $(3)$.

We summerize some basic properties of $\mathbb{H}(Y|X)$ under our definition $(7)$:

**Lemma 1.**    *(i) $0 \leq \mathbb{H}(Y|X) \leq \mathbb{H}(Y) \leq \log |\mathcal{Y}|$.*

*(ii) $\mathbb{H}(Y|X) = \mathbb{H}(Y)$ if and only if $X$ and $Y$ are independent.*

*(iii) $\mathbb{H}(Y|X) = 0$ if and only if $Y$ is a function of $X$. That is, there exist $|\mathcal{Y}|$ subsets of $\Omega$: $\Omega_1, \ldots, \Omega_{|\mathcal{Y}|}$ such that $P(X \in \Omega_y \cap \Omega_{y'}) = 0$ for any $y \neq y' \in \{1, \ldots, |\mathcal{Y}|\}$, $P(X \in \cup_{y=1}^{|\mathcal{Y}|} \Omega_y) = 1$ and $P(Y = y|X \in \Omega_y) = 1$ for any $y \in \{1, \ldots, |\mathcal{Y}|\}$.*

*Proof.*    (i)  This is trivial by $(7)$, $(9)$ and the fact that KL divergence is always nonnegative.

(ii)  By $(9)$, it is easy to get

$$\mathbb{H}(Y|X) = \mathbb{H}(Y) \iff X|Y = y \overset{d}{=\!=} X, \quad \forall y \in \{1, \ldots, |\mathcal{Y}|\}.$$
$$\iff P(X \in A, Y = y) = P(X \in A)P(Y = y), \text{for any}$$
$$\text{measurable set } A \text{ in } \mathcal{X}, \text{ any } y \in \{1, \ldots, |\mathcal{Y}|\}.$$
$$\iff X, Y \text{ are independent.}$$

(iii)  **Sufficiency:** If $Y$ is a function of $X$, then by assumptions, for any $y_1 \neq y_2$, we have

$$P(X \in \Omega_{y_1}, Y = y_2) \leq P(X \in \Omega_{y_1}) - P(X \in \Omega_{y_1}, Y = y_1) = 0.$$

Therefore, $p(x|Y = y_2) = 0$ for $x \in \Omega_{y_1}$(a.e.). Then for any $y \in \{1, \dots, |\mathcal{Y}|\}$, we have

$$\mathbb{KL}(X|Y = y) = \int_{\Omega_y} p(x|Y = y) \log \frac{p(x|Y = y)}{p_X(x)} dx \tag{10}$$

Notice that for $x \in \Omega_y$(a.e.), $p_X(x) = \sum_{y'=1}^{|\mathcal{Y}|} p_{y'} p(x|Y = y') = p_y p(x|Y = y)$, and thus

$$\mathbb{KL}(X|Y = y) = -\log p_y. \tag{11}$$

Then by (9), $\mathbb{H}(Y|X) = 0$.

**Necessity:** Assume $\mathbb{H}(Y|X) = 0$. Notice that $p_X(x) = \sum_{y=1}^{|\mathcal{Y}|} p_y p(x|Y = y), \forall x \in \Omega$. Therefore, for any $y \in \{1, \dots, |\mathcal{Y}|\}$,

$$\begin{aligned}
\mathbb{KL}(X|Y = y||X) &= \int_\Omega p(x|Y = y) \log \frac{p(x|Y = y)}{p_X(x)} dx \\
&\leq \int_\Omega p(x|Y = y) \log \frac{p(x|Y = y)}{p_y p(x|Y = y)} dx \\
&= -\log p_y.
\end{aligned} \tag{12}$$

Combing (9) and (12), we know if $\mathbb{H}(Y|X) = 0$, then

$$p_X(x) = p_y p(x|Y = y), \forall x \in \{x \in \Omega : p(x|Y = y) > 0\}(a.e.), \forall 1 \leq y \leq |\mathcal{Y}|. \tag{13}$$

Define $\Omega_y := \{x \in \Omega : p(x|Y = y) > 0\}$, for every $y \in \{1, \dots, |\mathcal{Y}|\}$. Without loss of generality, we assume $p_X(x) > 0$ for every $x \in \Omega$. (Otherwise, we can replace $\Omega$ by $\Omega \cap \{p_X(x) > 0\}$ and the following argument is still true.) Then we have

$$P(X \in \cup_{y=1}^{|\mathcal{Y}|} \Omega_y) = P(X \in \Omega) = 1.$$

By (13), we have

$$\begin{aligned}
P(X \in \Omega_y) &= \int_{\Omega_y} p_X(x) dx = \int_{\Omega_y} p_y p(x|Y = y) dx \\
&= p_y P(X \in \Omega_y | Y = y) = P(X \in \Omega_y, Y = y).
\end{aligned} \tag{14}$$

Therefore, $P(Y = y | X \in \Omega_y) = 1$. Lastly, we show $P(X \in \Omega_y \cap \Omega_{y'}) = 0$ for any $y \neq y'$. This is immediate if we notice that by replacing $\Omega_y$ in (14) with $\Omega_y \cap \Omega_{y'}$, we can have

$$P(X \in \Omega_y \cap \Omega_{y'}) = P(X \in \Omega_y \cap \Omega_{y'}, Y = y).$$

Changing the postion of $y$ and $y'$, we have

$$P(X \in \Omega_y \cap \Omega_{y'}) = P(X \in \Omega_y \cap \Omega_{y'}, Y = y').$$

Thus, $P(X \in \Omega_y \cap \Omega_{y'}) = 0$.

Before the end of the proof, we want to add a remark that the condition $P(X \in \Omega_y \cap \Omega_{y'}) = 0$ is actually just a consequence of $P(Y = y | X \in \Omega_y) = 1, \forall y$. We add it explicitly in the conditions to make it clear that $\Omega_1, \dots, \Omega_{|\mathcal{Y}|}$ is a partition of $\Omega$ according to the value of $Y$.

$\square$

## B   PROOF OF FANO'S INEQUALITY FOR CONTINUOUS FEATURES

We see the above essential properties for $\mathbb{H}(Y|X)$ have been preserved for the mixture case. Furthermore, we prove Fano's inequality still holds for our case. Let us restate Theorem 1 here:

**Theorem 2.** *Let $\mathcal{P}_e$ be the minimum error probability, i.e.*

$$\mathcal{P}_e = \inf_f P(f(X) \neq Y)$$

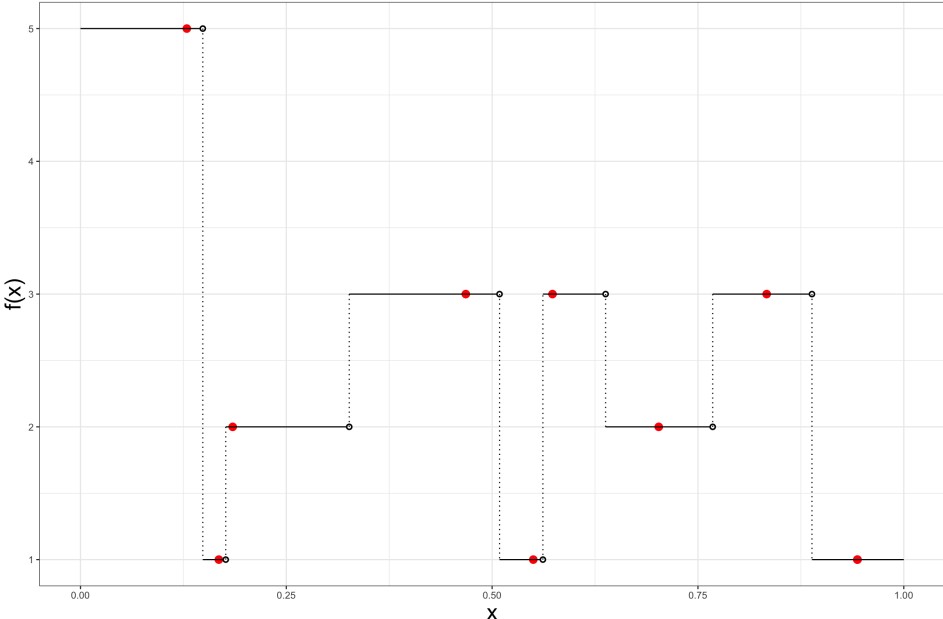

Figure 4: An example of $f$ with Smooth Discretization Property given the data and classifier. Red points represent $\{x_i, f_0(x_i)\}_{i=1}^{10}$, $x_i \sim U[0,1]$, $f_0$ is some given classifier in practice.

*where f is any "continuous" estimator of Y based on the observation X. That is, f is any function satisfying that for any $y = 1, \ldots, |\mathcal{Y}|$, almost every $x \in \Omega$, (x a.e. in $\Omega$)*

$$f(x) = y \iff \exists \delta > 0 \quad s.t. \quad \forall \tilde{x} \in B_\delta(x), \quad f(\tilde{x}) = y,$$

*where $B_\delta(x) := \{\tilde{x} \in \Omega : \|\tilde{x} - x\|_2 < \delta\}$ is a $\delta-$neighborhood of x in $\mathcal{X}$. Then we have*

$$\mathbb{H}(\mathcal{P}_e) + \mathcal{P}_e \log(|\mathcal{Y}| - 1) \geq \mathbb{H}(Y|X). \tag{15}$$

*Here, $\mathbb{H}(\mathcal{P}_e) := -\mathcal{P}_e \log \mathcal{P}_e - (1 - \mathcal{P}_e) \log(1 - \mathcal{P}_e)$.*

*Proof.* We basically follow the idea in the classical proof in Cover & Thomas (2012), although many details need to be dealt with carefully under our definition. Assume there exists a function $f$ achieves the minimum error probability, i.e. $P(f(X) \neq Y) = \mathcal{P}_e$(If such $f$ doesn't exist, we can always find some $f$ such that $P(f(X) \neq Y) \leq \mathcal{P}_e + \epsilon$ for some little $\epsilon > 0$, then let $\epsilon \to 0$ we can get the same result). Define an error random variable,

$$E = \begin{cases} 1, & \text{if} \quad f(X) \neq Y \\ 0, & \text{if} \quad f(X) = Y. \end{cases} \tag{16}$$

Under our definition (7), we have (in what follows, we use the notation $Y = i$ for $i = 1, \ldots, |\mathcal{Y}|$ instead of $Y = y$ to remind the reader that $Y$ is discrete.)

$$H(E, Y|X) := \mathbb{E}_X \left[ -\sum_{i=1}^{|\mathcal{Y}|} \sum_{j=0}^{1} \mathbb{P}(Y = i, E = j|X) \log \mathbb{P}(Y = i, E = j|X) \right]. \tag{17}$$

It is easy to check that

$$\mathbb{P}(Y = i, E = j|X) = \frac{P(Y = i, E = j)p(X|Y = i, E = j)}{p_X(X)}, \tag{18}$$

where $p(x|Y = i, E = j)$ is the conditional density function of $X$ given $Y = i, E = j$. Similarly, one may check that [3] [4]

$$\mathbb{P}(E = j|Y = i, X) = \frac{P(Y = i, E = j)p(X|Y = i, E = j)}{P(Y = i)p(X|Y = i)} \tag{19}$$

Combining the above two equations and (8), we have

$$\mathbb{P}(Y = i, E = j|X) = \mathbb{P}(Y = i|X)\mathbb{P}(E = j|Y = i, X). \tag{20}$$

Plug it in (17), and use the notation

$$\mathbb{H}(E|Y, X) := \mathbb{E}_X \left[ -\sum_{i=1}^{|\mathcal{Y}|} \sum_{j=0}^{1} \mathbb{P}(Y = i, E = j|X) \log \mathbb{P}(E = j|Y = i, X) \right], \tag{21}$$

we get

$$\mathbb{H}(E, Y|X) = \mathbb{H}(Y|X) + \mathbb{H}(E|Y, X) \tag{22}$$

For any $x \in \Omega, i = 1, \ldots, |\mathcal{Y}|$, we know $\exists \delta > 0$, such that

$$P(X \in B_\delta(x), Y = i, E = 0) = \begin{cases} P(X \in B_\delta(x), Y = i), & \text{if } f(x) = i \\ 0, & \text{if } f(x) \neq i. \end{cases} \tag{23}$$

Let $\delta \to 0$, we have $p(x|Y = i, E = 0)P(Y = i, E = 0) = p(x|Y = i)P(Y = i)$ or $0$ for any $x \in \Omega$ and $1 \leq i \leq m$. Therefore, by (19), we know $\mathbb{P}(E = 0|Y = i, X)$ is always $0$ or $1$ for any $1 \leq i \leq |\mathcal{Y}|$. So is $\mathbb{P}(E = 1|Y = i, X)$. Then by (21), we have

$$\mathbb{H}(E|Y, X) = 0. \tag{24}$$

Similar to (22), we have

$$\mathbb{H}(E, Y|X) = \mathbb{H}(E|X) + \mathbb{H}(Y|E, X), \tag{25}$$

where

$$\mathbb{H}(E|X) := \mathbb{E}_X \left[ -\sum_{j=0}^{1} \mathbb{P}(E = j|X) \log \mathbb{P}(E = j|X) \right] \tag{26}$$

and

$$\mathbb{H}(Y|E, X) := \mathbb{E}_X \left[ -\sum_{i=1}^{|\mathcal{Y}|} \sum_{j=0}^{1} \mathbb{P}(Y = i, E = j|X) \log \mathbb{P}(Y = i|E = j, X) \right]. \tag{27}$$

Notice that $\mathbb{H}(E|X) \leq \mathbb{H}(E) = \mathbb{H}(\mathcal{P}_e)$, it suffices to show

$$\mathbb{H}(Y|E, X) \leq \mathcal{P}_e \log(|\mathcal{Y}| - 1). \tag{28}$$

Similar to (23), we know for any $x \in \Omega, i = 1, \ldots, |\mathcal{Y}|, \exists \delta > 0$, such that

$$P(X \in B_\delta(x), Y = i, E = 0) = \begin{cases} P(X \in B_\delta(x), E = 0), & \text{if } f(x) = i \\ 0, & \text{if } f(x) \neq i. \end{cases} \tag{29}$$

Then we get $\mathbb{P}(Y = i|E = 0, X)$ is always $0$ or $1$ for any $i = 1, \ldots, |\mathcal{Y}|$. Therefore,

$$\mathbb{H}(Y|E = 0, X) := \mathbb{E}_X \left[ -\sum_{i=1}^{|\mathcal{Y}|} \mathbb{P}(Y = i, E = 0|X) \log \mathbb{P}(Y = i|E = 0, X) \right] = 0. \tag{30}$$

---

[3] Strictly speaking, by the definition of conditional expectation or probability, we should have $\mathbb{P}(E = j|Y = i, X) = \frac{P(Y=i,E=j)p(X|Y=i,E=j)}{P(Y=i)p(X|Y=i)}\mathbf{1}(Y = i) + \frac{P(Y=i,E=j)p(X|Y=i,E=j)}{P(Y\neq i)p(X|Y\neq i)}\mathbf{1}(Y \neq i)$.

[4] For the consistency of the notations, we still use $\mathbb{P}(E = j|Y = i, X)$ to denote the right side of (19). So actually, one should understand $\mathbb{P}(E = j|Y = i, X)$ as $\mathbb{P}(E = j|Y = i, X)\mathbf{1}(Y = i)$ in this context. Similar for $\mathbb{P}(Y = i|E = j, X)$ below.

For the case $E = 1$, we have

$$P(X \in B_\delta(x), Y = i, E = 1) = \begin{cases} P(X \in B_\delta(x), Y = i), & \text{if} \quad f(x) \neq i \\ 0, & \text{if} \quad f(x) = i. \end{cases} \tag{31}$$

Thus, we have

$$\mathbb{P}(Y = i | E = 1, X) = \frac{P(Y = i, E = 1) p(X | Y = i, E = 1)}{P(E = 1) p(X | E = 1)} \tag{32}$$

$$= \begin{cases} \frac{P(Y=i) p(X|Y=i)}{P(E=1) p(X|E=1)}, & \text{if} \quad f(X) \neq i \\ 0, & \text{if} \quad f(X) = i. \end{cases} \tag{33}$$

Similar to $(20)$, we have

$$\mathbb{P}(Y = i, E = 1 | X) = \mathbb{P}(E = 1 | X) \mathbb{P}(Y = i | E = 1, X). \tag{34}$$

Plug in the above two equations to $\mathbb{H}(Y | E = 1, X)$, we get

$$\mathbb{H}(Y | E = 1, X) = \mathbb{E}_X \left[ -\sum_{i=1}^{|\mathcal{Y}|} \mathbb{P}(E = 1 | X) \mathbb{P}(Y = i | E = 1, X) \log \mathbb{P}(Y = i | E = 1, X) \right]$$

$$= \mathbb{E}_X \left[ \mathbb{P}(E = 1 | X) \left( -\sum_{i \neq f(X)} \mathbb{P}(Y = i | E = 1, X) \log \mathbb{P}(Y = i | E = 1, X) \right) \right]$$

$$\leq \mathbb{E}_X \left[ \mathbb{P}(E = 1 | X) \log(|\mathcal{Y}| - 1) \right]$$

$$= \mathcal{P}_e \log(|\mathcal{Y}| - 1).$$

Therefore,

$$\mathbb{H}(Y | E, X) = \mathbb{H}(Y | E = 0, X) + \mathbb{H}(Y | E = 1, X) \leq \mathcal{P}_e \log(|\mathcal{Y}| - 1).$$

$\square$

## C    MODEL COMPLEXITY EXPERIMENT

Equation (6) suggests when the model class is larger the gap between DIME and the true lower bound on $\mathcal{P}_e$ becomes smaller. This is verified in our experiment shown in the middle of Figure (5), where we use neural networks with three hidden layers of varying number of neurons to calculate DIME. In the experiment we try 32, 64, 128, 256, 512, 1024, and 2048 neurons for each hidden layer. As the number of hidden neurons grows, DIME roughly decreases as expected. Still on CIFAR-10 the decrease is pretty monotonic. For simple datasets such as MNIST, Fashion MNIST and EMNIST (byletters) the numbers got stabilized pretty quickly.

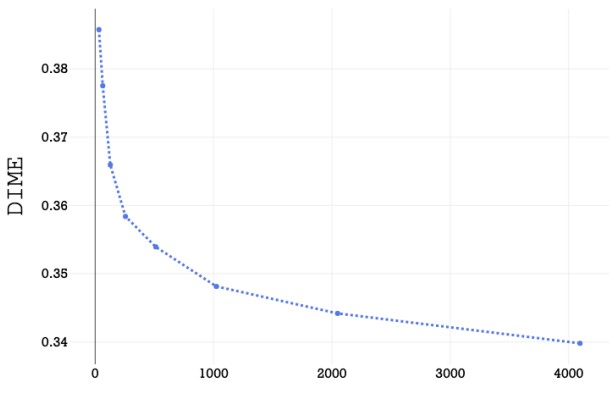

(a) DIME decreases on CIFAR-10 as the model size grows.

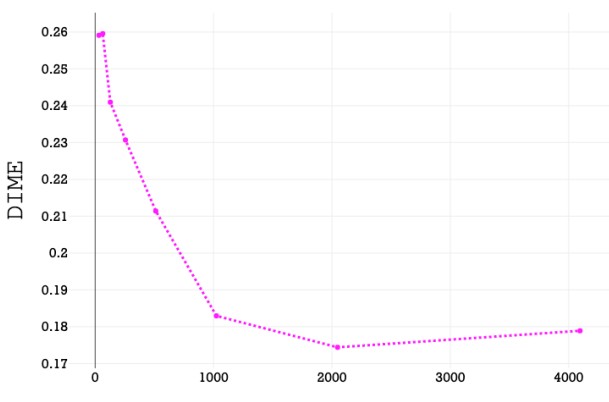

(b) DIME decreases on SVHN as the model size grows.

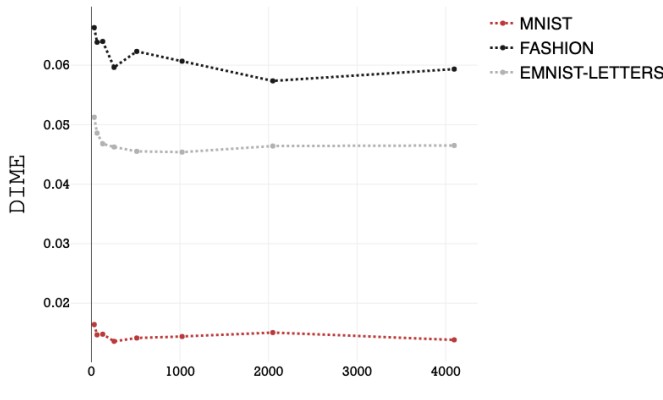

(c) DIME decreases on MNIST, Fashion MNIST, and EMNIST(byletters) as the model size grows.

Figure 5: DIME as the model size increases.

