# OpenReview forum: "DIME: AN INFORMATION-THEORETIC DIFFICULTY MEASURE FOR AI DATASETS"
_ICLR.cc/2020/Conference — Reject_

### Official Review · AnonReviewer1 · 2019-10-23
**Official Blind Review #1**

**Rating:** 3

**Review:**

So this paper is interesting. It's sort of pursuing a similar path as recent works that use neural networks to evaluate (e.g., inception score, FID), notably those that optimize some lower bound of an information measure (e.g., MINE). In this case, the setting is "datasets", and the thing they are trying to quantify is the difficulty of the dataset as expressed by a lower bound to the lowest possible probability of the 0-1 error, which they should is related to the conditional entropy of the underlying input / label random variables (which makes sense). This direction seems very useful, and using neural network optimization to attempt to crack defining "dataset complexity" or "difficulty" seems a worthwhile venture.

That said, I have some (potentially serious) concerns about some of the assumptions and approaches that may harm the validity of the proposal / results.

1) The smooth discretization property assumption seems to directly contradict results from adversarial examples, as these examples precisely a consequence of lacking smoothness. It seems that some constraints on the family of f are necessary to ensure this property (e.g., Lipschitz continuous using weight clipping, gradient penalty, spectral norm, etc).

2) The MINE estimator you use is asymptotically unbiased, and represents a *upper bound* of a lower bound of the KL when the number of samples is "small". The bound found in f-GAN is unbiased, yet tends not to perform well. I would also consider estimators found in [1] or use the bias-correcting found in MINE.

3) I feel like the model-agnostic claims are weak, due to the use of a particular family of functions used in the estimation part. Inductive bias must play a part in this, and some datasets "difficulty" is intimately connected to the class of functions we optimize our classifiers over. Some of these datasets may work better with some inductive biases, so it would be worth also looking at scores using convnets / LSTMs / transformers.

4) It might be better to relate "difficult" to real measures found in the literature. For example, one could remove examples known to be misclassified, etc. It would be good to see that removing these also lowers the DIME score.

Other comments:
Eq 2: m mysteriously appears
page 6, first paragraph: not normalizing the input worries me quite a bit. Did you show that the score is independent of normalization? This might have real consequences in conjunction with weight initialization, learning rates, etc.
Figure 2: it is good to see that label corruption correctly correlates with DIME score.
Table 1: it would be good to report a correlation between DIME and SOTA error.

Full disclosure, I did not go through the proofs in the appendix.

[1] On Variational Bounds of Mutual Information

Update:
Thank you for the responses to my questions. So since MINE is a central component to this work, I think it's value hinges on ensuring that the estimator bounds the thing that you're saying it should. The bias might not be a practical problem as far as optimizing or using this thing, but it does throw some of the numbers into question. Bias correcting or using / comparing to another optimizer is essential.

In addition, the inductive bias issue is still an important one, and I would expect this to be at least addressed in this type of work. The fact that your other models didn't work is an important point: why and what are the ingredients for a successful model?

**Experience Assessment:**

I have read many papers in this area.

**Review Assessment: Checking Correctness Of Derivations And Theory:**

I assessed the sensibility of the derivations and theory.

**Review Assessment: Checking Correctness Of Experiments:**

I assessed the sensibility of the experiments.

**Review Assessment: Thoroughness In Paper Reading:**

I read the paper at least twice and used my best judgement in assessing the paper.

---

> ### Author Response · Authors · 2019-11-15
> **Responses**
>
> Thank you for your feedback.
>
> Q. On the smooth discretization property assumption:
>
> A: Thanks for pointing this out.  We use the smooth discretization property assumption in our proof purely for technical reasons. We recently discovered a different approach to this proof in the book “Introduction to nonparametric estimation (Tsybakov, 2009)” (Lemma 2.10), which allows us to drop this assumption. We will modify our draft to update the proof.
>
> Q. “The MINE estimator you use is asymptotically unbiased, …”
> A: Thanks for the suggestion. The MINE estimator is indeed biased in the finite sample Monte Carlo estimation due to the log expectation term. Bias-correcting for MINE should be a promising approach for future work in this area.
>
>
> Q. “the model-agnostic claims are weak”
>
> A: As we emphasized in the Introduction of our paper,  our claim on model-agnosticity is only limited to the Fano’s bound and KL estimation if all the functions are considered in the optimization. In contrast, our empirical estimator is still model-dependent, since we use an MLP  to estimate the conditional entropy. In fact, we also tried other nonparametric estimators, such as knn or random projection based estimators, but they did not work as well as the neural networks.  We agree it would be worthwhile to look at the DIME using convnets/lstms/transformers. Our results take the first step in this direction, by showing that a standard MLP-based estimator obtains a reasonable upper-bound and ordering on dataset difficulty.
>
>
> Q. “one could remove examples known to be misclassified, etc. It would be good to see that removing these also lowers the DIME score.”
>
> A:  This is a great suggestion. We will add an experiment in the updated version, where we will compute the DIME score for datasets after removing misclassified samples by SOTA models.
>
>
> Q: “Did you show that the score is independent of normalization?“
>
> A: We perform no normalization of data except scaling the input from 0-255 to 0-1.  The estimation of KL between p(x|y_1) and p(x) should remain invariant to linear transformations. Another reason we did not do any normalization other than scaling is some normalization procedures may cause information loss. In fact, we had similar concerns about the effect of lack of normalization on  hyperparameters. However, we did not find this to be an issue in our experiments. Using stock initializations from Pytorch with learning rate of 0.05 for all the image datasets  (except TinyImagenet with a learning rate of 0.1) and a learning rate of  0.5 for text datasets worked well.

---

### Official Review · AnonReviewer4 · 2019-10-23
**Official Blind Review #4**

**Rating:** 1

**Review:**

This paper suggests a measure for the inherent difficulty of datasets: DIME.  Fano's inequality gives
a lower bound on the best possible predictor in terms of the conditional entropy of the labels
given the input.  The paper uses a MINE style estimator for this conditional entropy to give dataset
complexity measures.

The idea of using Fano's inequality and tight estimates of conditional entropy to give us a sense of how good we are doing on different datasets is a good one.

Howver, I believe this paper should be rejected. While the idea of using Fano's inequality to give a complexity score for datasets is interesting, this paper does not provide what appears to be a close measure of this conditional entropy.

Any fit model's log loss also provides a variational upper bound on the conditional entropy due to the positivity of KL:
$$ \int dx\, p(x) \int dy\, p(y|x) \log \frac{p(y|x)}{q(y|x)}  \geq 0   \implies  \int dx\, p(x) \int dy\, p(y) p(y|x) \log p(y|x) \geq  \int dx\, p(x) \int dy\, p(y|x) \log q(y|x)  \implies H(Y|X) \leq \mathbb{E}_{p(y,x)}[-\log q(y|x)]$$
The log loss of any model provides a variational upper bound on the conditional entropy H(Y|X).  In table 1, for all but EMNIST there exist models that give tighter upper bounds on the conditional entropy than DIME does.  While in principle an independent estimate of the conditional entropy might provide a useful signal that there is room to go in terms of the classification task (as it seems to do here in the case of EMNIST), in general the estimates provided by their MINE style estimator seem to not perform well.

Additionally, MINE does not provide a valid bound on the conditional entropy, (see e.g. "On variational bounds of mutual information" arXiv:1905.06922) as it when used as suggested it requires taking a monte carlo expectation inside a negative log.  A monte carlo estimate of a log expectation provides a stochastic lower bound of the log expectation (by Jensen's) so provides a stochastic upper bound of the proposed lower bound on KL, breaking the bound.  Couple this with the observation that MINE provides very high variance estimates of mutual information, in particular when the values are large, along with the general failure of the estimates in Table I and the specific combination of using MINE to give useful upper bounds on conditional entropy seems not to work well in practice, even on datasets with rather small expected conditional entropy (e.g. MNIST).  I suggest leveraging the known marginal in this instance to use an IWHVI style mutual information lower bound, as in
http://artem.sobolev.name/posts/2019-08-10-thoughts-on-mutual-information-more-estimators.html

The paper extends Fano's inequality to the mixed discrete-continuous case.  I always thought Fano's inequality would just carry over to the mixed discrete-continuous case because it doesn't rely at all on the cardinality of the input (X in this case).  Shouldn't a simple sup over all finite partitions of X give a natural extension (as it does to mutual information and conditional entropy in Cover & Thomas).  While there is utility in a careful extension or even clarifying note on the role of Fano's inequality in the mixed case, it's not clear it requires an entire paper, perhaps a note on the arxiv would help others who were interested in the extension.

In general, it doesn't seem as though the suggested method provides utility in giving better estimates of the difficulty of datasets than can already be gotten from our observed model log losses.  The MINE estimator is flawed and using it to estimate DIME is likely (and demonstrated in the paper) to give worse estimates than just training a conditional discriminative model and comparable cost, so I have to vote to reject the paper in its current form.

I do still think it is an interesting idea to try to assess whether or not we are within spitting distance of the optimal performance we could expect on our datasets.  To provide a proper lower bound would require a lower bound on the conditional entropy, for which I'm not sure of any general purpose results that could be leveraged, but in these settings it's not too dangerous to assume we have access to at least a known marginal over the labels and there is some hope that a decent lower bound on the conditional entropy might be able to be formed in this case.   For upper bounds, I really believe just the log loss of the best model the community has garnered is going to provide the tightest upper bound in almost all cases.   Could we still use this to figure out whether or not we have room to go?  While I have observed that in general for most models there isn't the best correlation between log loss and accuracy, the result of Fano's inequality suggests that when we really nail a dataset we might expect that the log loss provides a very tight upper bound on the true conditional entropy and the observed error rates provides a tight upper bound to the optimal Fano rate.  This suggest that it might be useful to collect paired results of the log loss and error from a whole slew of models and scatter the log loss versus ( H(e) + P(e) log (|Y|-1) ).   Then as just a sort of visual test as to whether the community is within spitting distance of the true discriminative density would be whether the best models start to show a linear relationship between their log loss and the transformed error rate.

**Experience Assessment:**

I have published in this field for several years.

**Review Assessment: Checking Correctness Of Derivations And Theory:**

I assessed the sensibility of the derivations and theory.

**Review Assessment: Checking Correctness Of Experiments:**

I assessed the sensibility of the experiments.

**Review Assessment: Thoroughness In Paper Reading:**

I read the paper at least twice and used my best judgement in assessing the paper.

---

> ### Author Response · Authors · 2019-11-15
> **Responses**
>
> Thank you very much for your detailed feedback.
>
> Q: Any fit model’s log loss also provides a variational upper bound…
>
> A: While it is true that any fit model’s log loss gives an upper bound, we are suggesting an  information-theoretical perspective to guide the dataset design and evaluation. Though both can provide an upper bound, the methods those algorithms are based on are fundamentally different. Moreover, our theoretical analysis shows a model-agnostic method, though the empirical estimator is not. Finally, our goal is to provide a method for comparing difficulty *across* datasets and modalities, while a typical model-fitting focuses on minimizing loss on a particular dataset. The neural networks in our methods, which maps the input to a scalar, are purely used for the estimation of the Kl divergence. While the fit model, which maps the input to a vector of softmax ``probabilities”, mainly shoot for a better prediction accuracy.
>
> Q: for all but EMNIST there exist models that give tighter upper bounds on the conditional entropy than DIME does
>
> A: Most of the data sets in Table 1 are pretty old. Since the community has been working on most of these datasets for years, the fit models are perform remarkably well, due to factors including test-set data leakage [Recht et al., 2018]. EMNIST is a good example of a less-explored dataset, on which our method provides a better bound than the state-of-the-art, even when using a basic multi-layer perceptron without any convolution layers and feature engineering. Moreover, even if the bound is not tight, the relative order across datasets is roughly aligned with the SOTA errors.
>
> Recht, B., Roelofs, R., Schmidt, L. and Shankar, V., 2018. Do CIFAR-10 classifiers generalize to CIFAR-10?. arXiv preprint arXiv:1806.00451.
>
> Q: “The MINE estimator is flawed and using it to estimate DIME is likely (and demonstrated in the paper) to give worse estimates”
>
> A: We agree that MINE estimators do have limitations when using Monte-Carlo estimations for the log expectation term. The MINE estimation is biased and a de-biasing process may be added. Even if we add a debiasing process,  our proposed procedure still remains as an upper bound of a lower bound, which “breaks the lower bound”. This is because: 1 we are trying to estimate a lower bound of the error probability indicated by Fano’s theorem. 2. We only consider  a small subset of the whole function space when estimating the KL divergence, which results in a higher conditional entropy.
>
> Unfortunately, in empirical estimation, several issues could break the assumptions in the analysis. For example, neural networks can almost fit arbitrary functions. As a result,  when they are  used to estimate the KL divergence (by Donsker’s representation) using finite training samples, they can return extremely large values if we do not use validation sets. This is not well studied in our mutual information estimation either, since we use the population entropy and mutual information in our analysis. There are also issues caused by local optimum,  learning rate, choice of optimizers, and initializations.
>
> Q: “I always thought Fano's inequality would just carry over to the mixed discrete-continuous case”
>
> A: While it may seem so, there is no trivial way to extend Fano’s inequality to the mixed discrete-continuous case.  We haven’t been able to come up with an approach following your suggestion by taking sup over all finite partitions of X. Following the proof in Cover & Thomas does not give us a full proof for this case. But, we found a proof on page 111 of book “Introduction to Nonparametric Estimation” by Alexandre B. Tsybakov during the rebuttal process,, which can be easily generalized to the continuous feature case. We will add that to our draft. The proof in the book is from a perspective that is very different compared to ours.
>
> Q: I suggest leveraging the known marginal in this instance to use an IWHVI style mutual information lower bound
>
> A: This is an interesting idea and can definitely lead to promising future work.
>
> Q: For upper bounds, I really believe just the log loss of the best model the community has garnered is going to provide the tightest upper bound in almost all cases.   Could we still use this to figure out whether or not we have room to go?
>
> A:  While a proper lower bound of conditional entropy is not easy and model log losses can provide tight upper bounds, it might take the community some time to converge on the right model architectures for new datasets.  In contrast, our metric can readily provide a metric for dataset difficulty.
>
> Q: “This suggest that it might be useful to collect paired results of the log loss and error from a whole slew of models and scatter the log loss versus ( H(e) + P(e) log (|Y|-1) ).”
>
> A: This is a great suggestion. We believe this could be a great future direction to investigate.

---

> > ### Comment · AnonReviewer4 · 2019-11-15
> > **Paper still seems misleading as written**
> >
> > Thank you for your thoughtful response.
> >
> > But it seems as though for the most part you agree with several of my criticisms.  Log loss of some model often provides a better approximation of the fano optimal error, and offers a true bound on the mutual information.  The MINE style estimator of the paper fails to provide a valid bound.  It additionally fails to demonstrate its utility as an estimator.  In particular, I would have liked to see a baseline comparison to attempting to optimize the log loss with a similar architecture as used to power the MINE style discriminator.
> >
> > The biggest problem I have is that it does seem like there are valid criticisms of the paper, but the current paper as written doesn't really highlight any of these.
> >
> > I feel I have to stick with a vote to reject as I feel as though the paper in its current form could be misleading to most readers.
> >
> > I do think there is an interesting idea at its core and I encourage the authors to keep at it, and look forward to improved versions of the paper.

---

### Official Review · AnonReviewer2 · 2019-10-23
**Official Blind Review #2**

**Rating:** 3

**Review:**

The paper proposes a measure of difficulty for datasets. Prior work in this space has often utilized certain indicators like the overlap of samples across different classes etc. [A] While this work defines a model-agnostic error as the measure of difficulty, which should encompass all possible indicators of error. Then, the paper provides a lower bound on this error which can be estimated using neural network [B]

I am inclined to accept (weak) this paper for the following reasons:
1. The paper extends Fano's inequality for the case of real-valued vectors and discrete labels, under the assumption of smoothness of the estimators.
2. The proposed approach for difficulty measure estimation is simple and clear and primarily based on [B].
3. The estimates seem to correlate well with the errors of state of the art models, particularly on sentiment and text classification dataset. On the image datasets, it is still reasonably well correlated.

Some suggestions for improvement:
1. Add prior work on measuring data complexity to the references, e.g. [A, C] etc. and add some details contrasting prior work with this paper
2. It would also be good to see some correlation numbers like Pearson correlation etc. between DIME and SOTA errors.

[A] Spectral Metric for Dataset Complexity Assessment, CVPR 2019
[B] Mutual Information Neural Estimation, ICML 2018
[C] Complexity Measures of Supervised Classification Problems, PAMI 2002

---
Update:

Thanking authors for all the thoughtful rebuttal made to other reviewers.

Based on the concerns raised by the other reviewers and looking through the comments, I am inclined to lower my scores to a weak reject. There is definitely quite a lot of good ideas in this paper and it might just be a matter of bulking up with more analysis at this point as suggested by other reviewers.

**Experience Assessment:**

I do not know much about this area.

**Review Assessment: Checking Correctness Of Derivations And Theory:**

I assessed the sensibility of the derivations and theory.

**Review Assessment: Checking Correctness Of Experiments:**

I assessed the sensibility of the experiments.

**Review Assessment: Thoroughness In Paper Reading:**

I read the paper at least twice and used my best judgement in assessing the paper.

---

> ### Author Response · Authors · 2019-11-15
> **Responses**
>
> Thank you for the comments and suggestions. We will add a discussion on the contrast between prior work and our approach as you suggest.We will also compute the Pearson correlation between DIME and SOTA errors  and add them to the results.

---

### Decision · Program_Chairs · 2019-12-19

**Decision:**

Reject

**Comment:**

This paper proposes a measure of inherent difficulty of datasets. While reviewers agree that there are good ideas in this paper that is worth pursuing, several concerns has been risen by reviewers, which are mostly acknowledged by the authors. We look forward to seeing an improved version of this paper soon!